# Investigation of the Pathogenic Mechanism of Ciprofloxacin in Aortic Aneurysm and Dissection by an Integrated Proteomics and Network Pharmacology Strategy

**DOI:** 10.3390/jcm12041270

**Published:** 2023-02-06

**Authors:** Zhaoran Chen, Jianqiang Wu, Wei Wang, Xiaoyue Tang, Lei Zhou, Yanze Lv, Yuehong Zheng

**Affiliations:** 1Department of Geriatrics, Beijing Friendship Hospital, Capital Medical University, Beijing 100050, China; 2State Key Laboratory of Complex Severe and Rare Disease, Department of Medical Research Center, Peking Union Medical College Hospital, Chinese Academy of Medical Sciences and Peking Union Medical College, Beijing 100730, China; 3Department of Vascular Surgery, Peking Union Medical College Hospital, Chinese Academy of Medical Sciences and Peking Union Medical College, Beijing 100730, China; 4Department of Vascular Surgery, Beijing Friendship Hospital, Capital Medical University, Beijing 100050, China

**Keywords:** aortic aneurysm and dissection, fluoroquinolones, proteomics, network pharmacology, molecular docking

## Abstract

Aortic aneurysm and dissection (AAD) is a life-threatening disease worldwide. Recently, fluoroquinolones have been reported to significantly increase the risk of AAD. This study aimed to investigate the potential functional mechanism and molecular targets of fluoroquinolones in relation to AAD by an integrated proteomic and network pharmacology strategy. A total of 1351 differentially expressed proteins were identified in human aortic vascular smooth muscle cells (VSMCs) after ciprofloxacin (CIP) stimulation. The functional analysis emphasized the important roles of metabolism, extracellular matrix homeostasis, mitochondrial damage, focal adhesion, and apoptosis in CIP-stimulated VSMCs. CIP targets were predicted with online databases and verified by molecular docking. Protein–protein interaction (PPI) analysis and module construction of the 34 potential CIP targets and 37 selected hub molecules after CIP stimulation identified four critical target proteins in the module: PARP1, RAC1, IGF1R and MKI67. Functional analysis of the PPI module showed that the MAPK signalling pathway, focal adhesion, apoptosis, regulation of actin cytoskeleton, and PI3K-Akt signalling pathway were significantly enriched. Our results will provide novel insights into the pathogenic mechanism of fluoroquinolones in aortic diseases.

## 1. Introduction

Aortic dissection and aneurysm (AAD) is a cardiovascular emergency involving the separation and/or dilation of the aortic wall, and with fatal consequences if therapy is not performed in time [1,2]. The critical histopathologic feature of AAD is medial degeneration characterized by vascular smooth muscle cell (VSMC) changes, extracellular matrix (ECM) degradation, and inflammation [3,4]. VSMCs are the most important cellular components of the aortic media and play a central role in establishing and maintaining aortic homeostasis. The aortic media is composed mainly of VSMCs and ECM, which jointly maintain the contractile function and structural stability of the aorta. Under physiological conditions, VSMCs can synthesize ECM components. However, when VSMCs are injured and stimulated, they secrete proinflammatory cytokines and matrix metalloproteinases (MMPs) that degrade the ECM of blood vessels and recruit inflammatory cells to aggravate the inflammatory response, thus destroying the aortic structure and function. Increasing amounts of evidence show that the loss and dysfunction of VSMCs can induce AAD formation [5]. However, the specific molecular mechanism of aortic wall degeneration has yet to be fully elucidated.

Fluoroquinolones (FNs) are among the most commonly used antibiotics in clinical practice and have been used to treat various infectious diseases, including infectious aortic aneurysms [6]. FNs, which are broad-spectrum antibiotics, have good tissue and cell penetration abilities. However, many studies have indicated that FNs are associated with an increased incidence of AAD. According to different population-based studies, FN use increases the risk of AAD with a hazard ratio from 1.31 to 2.43 [7,8,9,10]. Moreover, FN exposure was reported to be associated with an increased risk of adverse outcomes in AAD patients [11]. According to the Food and Drug Administration (FDA), FNs should not be used by patients with high-risk conditions, such as those with Marfan syndrome or aortic diseases [12]. In addition, ciprofloxacin (CIP), one of the most commonly used FNs, can increase susceptibility to AAD formation and aortic rupture in animal models [13]. Therefore, it is necessary to clarify the mechanism of FNs in aortic disease and to evaluate the risks of FNs.

Proteomics is a systems biology approach that can be used to characterize protein species on a large scale. Mass spectrometry (MS)-based proteomics techniques can simultaneously detect hundreds to thousands of proteins in samples. Recently, data-independent acquisition (DIA)-based MS has increased the sensitivity, quantitative accuracy and reproducibility of proteomics analyses. Proteomics is a promising technology for pathogenesis exploration and drug target identification [14].

Evidence has shown that loss and dysfunction of VSMCs is associated with the occurrence and development of AAD and an increased risk of AAD in FNs use. Thus, we hypothesized that CIP might affect the proliferation and survival of VSMCs by certain mechanisms. Consequently, we used a DIA MS-based proteomics method to analyse differentially expressed proteins (DEPs) in cultured VSMCs stimulated with CIP. We also sought to discover the critical molecules and pathways involved in the pathogenesis of AAD caused or aggravated by CIP using bioinformatics and network pharmacology methods. Our results will provide novel insights into the pathogenic mechanism of FNs in aortic diseases.

## 2. Materials and Methods

### 2.1. Cell Culture and Viability Assays

Human aortic smooth muscle cells (HASMCs) were purchased from ScienCell Research Laboratories (ScienCell, CA, USA). The HASMCs were cultured in DMEM-F12 medium (HyClone, UT, USA) with 10% foetal bovine serum in a humidified atmosphere containing 5% CO_2_ at 37 °C, and cells passaged 5 to 7 times were used in the experiments. Cell proliferation was measured using the Cell Counting Kit-8 (CCK-8) assay. VSMCs were seeded onto 96-well plates and stimulated with sterile saline or three concentrations of CIP (100 μg/mL, 200 μg/mL, 300 μg/mL) for 48 h. CCK-8 reagent was added to each well, and the cells were incubated for 2 h at 37 °C. The absorbance of the wells at a wavelength of 450 nm was measured with a microplate reader.

### 2.2. Sample Preparation

Cells were treated with 100 μg/mL CIP or sterile saline for 48 h. The cell samples were washed with cold phosphate-buffered saline to eliminate blood contamination and homogenized with protein lysis buffer (7 M urea, 1% protease inhibitor cocktail) on ice for 20 min. A supernatant was collected after centrifugation at 12,000× *g* for 15 min at 4 °C, and the protein concentrations were determined by Bradford assay.

The protein samples were prepared using the filter-aided sample preparation (FASP) method for tryptic digestion [15]. In brief, the proteins in the samples were reduced with 10 mM DTT at 37 °C for 1 h and alkylated with 25 mM IAA at room temperature in the dark for 30 min. Each sample was loaded into a 30 kDa ultracentrifugation filter and washed three times with 25 mM NH_4_HCO_3_. Then, the samples were digested with MS-grade trypsin at 37 °C overnight. The digested peptides were desalted on C18 columns (Oasis, Waters Corporation, Milford, MA, USA). Then, the desalted peptides were lyophilized by vacuum centrifugation and redissolved in 0.1% formic acid (FA). The peptide concentration in each sample was quantified by the BCA method. One microgram of each sample was used for LC–MS/MS analysis.

### 2.3. DIA-Based Proteomics Analysis

The proteomics analysis was conducted with an Orbitrap QE HF mass spectrometer coupled to an EASY-nLC 1200 UHPLC system (Thermo Fisher Scientific, Waltham, MA, USA). The peptides were loaded onto a trap column and separated on an analytical column (75 µm × 500 mm, Kyoto Monotech, Kyoto, Japan). Peptides were eluted using a 60 min gradient from 5 to 30% buffer B (0.1% FA in 99.9% acetonitrile) at a flow rate of 300 nl/min. Moreover, all samples were spiked with iRT peptides for alignment of retention times across the samples. The mass spectrometer was operated in DIA mode, and a full-scan MS spectrum (350–1500 *m*/*z*) was collected with a resolution of 120,000. The injection time was less than 100 ms, and the automatic gain control (AGC) target value was 3e6.

### 2.4. MS Data Processing and Analysis

All raw mass spectrometric data were imported into Spectronaut software (Biognosys AG, Schlieren, Switzerland) to generate the in-house-developed spectral library. The proteomics data were searched against the UniProt human database appended with the iRT sequence. The parent ion tolerance and fragment ion tolerance were set to 10 ppm and 0.05 Da, respectively. Not more than two missed cleavage sites for trypsin digestion were allowed. Carbamidomethylation of cysteines was considered the fixed modification, and oxidation of methionine was set as the variable modification. Then, the DIA raw files were searched against the spectral library. The cross-run normalization was based on the local regression. The Q value was set to 0.01 for proteomics data filtering. The summed peak areas in MS2 were used for peptide quantification. The gene expression matrix files with an official gene symbol were obtained from the DIA platform after normalization for DEP analysis. The thresholds for DEP identification were an absolute log2(fold-change) (log FC) value of > 1 and an adjusted *p* value (q) of < 0.05. The UniProt database was used for gene symbol standardization [16].

### 2.5. Gene Set Enrichment Analysis (GSEA)

GSEA was conducted using the GSEA website (http://www.broadinstitute.org/gsea, accessed on 3 July 2022) [17]. We chose to identify the most significant biological processes and Kyoto Encyclopedia of Genes and Genomes (KEGG) pathways between the CIP stimulation and normal saline control groups. The “c2.cp.kegg.v7.5.1.symbols.gmt” and “c5.go.v7.5.1.symbols.gmt” gene sets derived from the Molecular Signatures Database (MSigDB) were used as the reference gene sets, and the number of random sample permutations was set to 1000. The significance thresholds were a false discovery rate (FDR) < 0.25 and a *p* value <0.05.

### 2.6. Acquisition of AAD-Related Genes from the Comparative Toxicogenomics Database (CTD)

The CTD (http://ctdbase.org, accessed on 3 July 2022) provides manually curated information about chemical–gene/protein, chemical–disease, and gene–disease relationships [18]. Relevant genes were obtained by retrieving genes associated with the keywords “aortic aneurysm” (MeSH ID: D001014) and “aortic dissection” (MeSH ID: D000784). After merging the two gene lists and removing duplicates, the genes with more than 5 references were selected.

### 2.7. Functional Enrichment Analysis

The intersection of the DEP-encoding genes and AAD-related genes from the CTD was subjected to Gene Ontology (GO) and KEGG enrichment analyses using the DAVID Bioinformatic Resources database (https://david.ncifcrf.gov/, accessed on 3 July 2022) [19]. The terms and pathways with *p* < 0.05 were considered significantly enriched for the DEPs. The top 10 enriched GO terms (ranked by *p* value) for the upregulated and downregulated DEPs are shown. The identified KEGG pathways were visualized with the Cytoscape plugins ClueGO and CluePedia [20].

### 2.8. Protein–Protein Interaction (PPI) Network Construction and Identification of Key Modules and Hub Genes

The PPI network was constructed using the STRING database (http://string-db.org/, accessed on 3 July 2022) [21]. The overlapping DEPs were analysed with the STRING database with a confidence score fixed at 0.4 (medium confidence). Then, Cytoscape software (version 3.9.1, https://cytoscape.org/, accessed on 3 July 2022) was used to construct and visualize the PPI network of the overlapping DEPs [22]. Functional modules were identified with the Cytoscape plugin MCODE using the default parameters (degree cut-off = 2, node score cut-off = 0.2, K-core = 2 and max depth = 100) [23]. Then, another Cytoscape plugin, cytoHubba, was used to identify the hub genes [24]. In cytoHubba, several common algorithms (MCC, MNC, Degree, EPC, Closeness, Radiality and Stress) were used to evaluate and screen the hub genes.

### 2.9. Target Gene Prediction and Molecular Docking

The SDF format files for CIP were derived from the NCBI PubChem database (https://pubchem.ncbi.nlm.nih.gov/, accessed on 3 July 2022). Chem3D software (https://library.bath.ac.uk/chemistry-software/chem3d, accessed on 3 July 2022) was used to convert the SDF files to mol2 format, and the 2D structure was converted into a 3D structure. The potential target genes of CIP were acquired from PharmMapper (http://www.lilab-ecust.cn/pharmmapper/, accessed on 3 July 2022) [25] and the Swiss Target Prediction database (http://www.swisstargetprediction.ch/index.php, accessed on 3 July 2022) [26]. The UniProt database was used for comparison of target information and gene symbol information. The target genes were searched in the Protein Data Bank (PDB) database (https://www.rcsb.org, accessed on 3 July 2022), and their 3D protein conformations were acquired. Water molecule deletion and hydrogen addition were performed with PyMOL software for pre-treatment. AutoDockTools 1.5.7 software (http://autodock.scripps.edu/, accessed on 3 July 2022) was applied for molecular docking according to the original ligand coordinates, and the docking box was adjusted to include all protein structures. The docking results were obtained by running AutoDock Vina, which revealed the binding energies. These molecular docking results were then visualized by PyMOL software (https://pymol.org/, accessed on 3 July 2022). The study design and flow chart of this study are shown in Appendix A.

## 3. Results

### 3.1. Identification of DEPs under CIP Stimulation

The CCK-8 assay results showed that CIP significantly inhibited the proliferation of VSMCs in a dose-dependent manner (Figure 1A). A total of 3825 proteins were identified in the VSMCs using DIA quantitative proteomics analysis (Appendix A). Ultimately, 1351 proteins were found to be significantly differentially expressed (ratio ≥ 2, and adjusted *p* value < 0.05) in samples stimulated with CIP compared with normal saline-treated control samples (Appendix A). Among these DEPs, 796 were upregulated and 555 were downregulated. The expression patterns of these proteins are shown in Figure 1B. We selected the top 100 DEPs (50 upregulated, 50 downregulated) for cluster analysis (Figure 1C).

### 3.2. Gene Set Enrichment Analysis (GSEA)

GSEA was conducted to investigate the differences in enriched GO terms and KEGG pathways between CIP-stimulated and control samples. The enriched biological processes (BPs) indicated that metabolic processes such as the tricarboxylic acid cycle (TCA), nucleoside bisphosphate metabolic process and guanosine monophosphate metabolic process were upregulated, whereas ion binding, components of smooth muscle contraction and ECM components showed trends of downregulation, as shown in detail in Figure 2.

### 3.3. Proteins Related to CIP Stimulation and AAD

A search of the CTD database (keywords: aortic dissection and aortic aneurysm) revealed a total of 17,046 aortic dissection-related genes and 33,837 aortic aneurysm-related genes. After merging and duplicate removal, 2088 AAD-related genes were ultimately selected according to previous studies with more than five references (Appendix A). Overlapping of the coding genes of the 1351 CIP-stimulated DEPs and 2088 CTD-based genes revealed 211 overlapping differentially expressed genes (DEGs) for further analysis (Figure 3A,B).

### 3.4. Analysis of the Functional Characteristics of the Overlapping DEGs

GO term and KEGG pathway enrichment analyses were conducted to analyse the biological functions and pathways involving those 211 overlapping DEGs. Regarding BPs, the cellular response to mechanical stimulus and cellular response to cadmium ion BP terms were significantly enriched for the upregulated DEGs, and the regulation of apoptosis term was significantly enriched for both the upregulated and downregulated DEGs (Figure 3C,D). In the cellular component (CC) category, the cytosol, extracellular space, mitochondrion, and focal adhesion GO terms showed significant changes (Figure 3E,F). In the molecular function (MF) category, RNA binding, enzyme binding and ATP binding were enriched for the upregulated DEGs, while integrin binding, actin binding and protease binding were enriched for the downregulated DEGs (Figure 3G,H). In addition, KEGG analysis results showed that the overlapping DEGs were significantly enriched in apoptosis, the AGE-RAGE signalling pathway, focal adhesion, the relaxin signalling pathway, reactive oxygen species, the PI3K-Akt signalling pathway, the Ras signalling pathway, the TNF signalling pathway, the GnRH signalling pathway, and tight junctions (Figure 4).

### 3.5. PPI Network Construction, Module Analysis, and Hub Gene Identification

The PPI network of the overlapping DEGs with combined scores > 0.4 was obtained from the STRING database and then visualized using Cytoscape software (https://cytoscape.org/, accessed on 3 July 2022). This PPI network contained 200 nodes and 1207 edges (Figure 5A). The top four closely connected gene modules were identified with MCODE, and the results are presented in Figure 5B–E. With another plugin of Cytoscape, cytoHubba, we determined the top 20 hub genes via seven main algorithms, and the results are shown as stacked bar plots (Figure 6). Among these genes, GAPDH, FGF2, EGFR, ALB, JUN, MAPK3, RHOA, CASP3, and CDC42 were identified as significant by all seven algorithms.

### 3.6. Proteins Related to CIP Stimulation and AAD

The targets of CIP were predicted with the databases PharmMapper and Swiss Target Prediction. Ultimately, 456 targets were obtained (Appendix A). Overlap of these targets with CTD-based AAD-related genes identified 34 overlapping target genes (Figure 7A). Then, to verify and explore the interaction mode between ligand molecules and receptor protein macromolecules, molecular docking was performed to calculate the binding energy between CIP and these overlapping target proteins to predict the binding affinity (Table 1). The proteins with binding energies less than −8 kcal/mol are shown in Figure 7B–K. The low binding energies indicated a stable conformation and suggested that these molecules spontaneously bind to CIP. To identify the potential mechanism of CIP in AAD, PPI analysis was performed on the 34 potential CIP targets and the hub proteins after CIP stimulation (listed in Figure 6). A new PPI network was constructed (Figure 8A), and a closely connected gene module was calculated from this network with the plugin MCODE (Figure 8B). Finally, four critical targets of CIP were identified in the module: PARP1, RAC1, IGF1R and MKI67. Functional analysis of this PPI module showed that the MAPK signalling pathway, focal adhesion, apoptosis, regulation of actin cytoskeleton and PI3K-Akt signalling pathway were significantly enriched.

## 4. Discussion

The present study conducted a DIA-based proteomics analysis of CIP-treated VSMCs and identified 1351 DEPs. The GSEA results showed that proteins related to some metabolic processes were significantly upregulated, whereas proteins related to ion binding, smooth muscle contraction components and ECM components were significantly downregulated after CIP treatment. After the DEPs overlapped with the AAD-related genes, functional analysis emphasized the important roles of metabolism, ECM homeostasis, mitochondrial damage, focal adhesion and apoptosis in CIP-stimulated VSMCs. Construction and analysis of a PPI network identified several modules and hub proteins. Through database prediction, molecular docking and PPI module analysis, we identified four critical molecules (PARP1, RAC1, IGF1R and MKI67) that are AAD-related and can spontaneously bind to CIP and regulate the function of VSMCs. These proteins are potentially crucial in CIP stimulation and AAD pathogenesis.

AAD is a potentially life-threatening disease, and its pathogenesis is not well understood. Clinical reports have indicated that FNs are associated with adverse vascular effects, such as AAD. A previous study showed that CIP increases the incidence and disease severity of AAD. The results suggested that CIP could induce degradation of the ECM and injury to VSMCs [13]. The ECM is one of the most important structural components of the aorta and provides elasticity and tensile strength to blood vessel walls. Increased MMP expression and activation can increase ECM destruction [27,28]. Notably, it has been reported that CIP can increase the expression of MMP-2 and MMP-9 in the aortic wall [13], cornea [29], tendon cells and tissues [30,31] and fibroblasts [32]. In this study, we failed to identify MMP-2 and MMP-9 expression in CIP-stimulated VSMCs, possibly because these MMPs are secreted [33] and are, therefore, difficult to detect by MS. However, we found that MMP14, a membrane-type MMP, was significantly downregulated in CIP-stimulated VSMCs. Nevertheless, the expression trend of MMP14 in the current study is different from a previous finding in thoracic aortic aneurysms [27] but consistent with findings in CIP-stimulated HepG2 liver cells in a Gene Expression Omnibus (GEO) dataset (GSE9166) [34]. Lysyl oxidase (LOX) plays a critical role in maintaining aortic wall integrity by participating in the assembly and stabilization of elastic fibres. Similar to previous studies, we found reduced production of LOX in CIP-treated VSMCs.

Loss of VSMC is another prominent pathological characteristic of AAD, and progressive VSMC loss aggravates AAD and ultimately results in aortic rupture [35,36,37]. CIP has been shown to induce apoptosis in lung, colon and prostate cancer cells [38,39,40]. In the current study, both GSEA and functional enrichment analysis emphasized the importance of VSMC apoptosis under CIP stimulation. Our results also showed functional changes in the AGE-RAGE signalling and focal adhesion pathways. In PPI analysis, we identified the hub genes via different algorithms and seven genes were identified as significant by all the algorithms. Most of these genes were central regulators in the MAPK signalling pathway, the Rap1 signalling pathway, focal adhesion, and the regulation of the actin cytoskeleton. These results might indicate the pathways that mediate CIP-induced dysfunction of VSMCs.

In this study, we identified four critical proteins that were predicted to be CIP targets: PARP1, RAC1, IGF1R and MKI67. Poly (ADP-ribose) polymerase 1 (PARP) is a DNA damage sensor and signalling molecule that can regulate MMPs, ROS, MAPK and NFκB [41,42]. Genetic deletion of PARP1 has been found to prevent abdominal aortic aneurysm formation in an animal model [43]. Ras-related C3 botulinum toxin substrate 1 (RAC1) encodes a small GTPase that can bind to effector proteins to regulate various cellular responses [44]. The activation of RAC1 can promote the migration and proliferation of VSMCs [45]. In contrast, the inhibition of RAC1 can significantly inhibit the activation of NF-κB and lead to decreased expression of MMP-9, MCP-2 and CXCL5 in abdominal aortic aneurysm tissue [46]. A recent study identified IGF1R as a critical gene in aortic dissection [47]. The binding of IGF1R can activate the MAPK or PI3K-Akt pathway to regulate the cellular proliferation or apoptosis of VSMCs. The primary function of MKI67 is to maintain mitotic chromosome separation by forming a steric and electrostatic charge barrier [48]. MKI67 also plays a key role in cell proliferation. Our molecular docking results indicated that CIP has superior affinity for these proteins, which may be potential therapeutic targets for AAD.

FNs have broad-spectrum antimicrobial activity and high penetration into tissues. Although there is increasing evidence of their cardiovascular side effects, FNs are still important tools in clinical practice. It is therefore necessary to clarify the mechanism underlying the cytotoxicity of FNs in AAD and find potential targets for aortic events. Although previous studies have separately explored the hub genes of AAD [49,50,51,52], few studies have investigated the common molecular mechanism between FNs and AAD via advanced bioinformatics approaches. The promising approach we used in this study, proteomics, is a powerful tool for pathogenesis exploration and drug target identification. Meanwhile, novel DIA-based proteomics technology promises the robust and accurate quantification of proteins, and is increasingly being applied in cardiovascular disease research [53,54]. Our findings have important clinical implications, but several limitations of this study need to be mentioned. First, although VSMCs remain the most important cellular components in the pathological process of AAD, other cells, such as vascular endotheliocytes and macrophages, also participate in the development of AAD. This study only focused on VSMCs, and the conclusion may not be extended to other cellular components. Second, consistent with previous studies that explored the effect of ciprofloxacin on cellular function in vitro [55,56], the concentrations of CIP used in our study were 100-300 μg/mL. However, the peak concentration of CIP in human plasma is approximately 1.26 μg/mL after administration of a single oral dose of 250 mg [57]. The gap between the drug concentration used in the in vitro experiment and the actual blood concentration may affect the interpretation of the current results. Third, all the data we analysed are from in vitro experiments and online databases, and confirmation via in vivo experiments will be needed in the future. In summary, our results will provide novel insights into the pathogenic mechanism of FNs in aortic diseases.

## Figures and Tables

**Figure 1 jcm-12-01270-f001:**
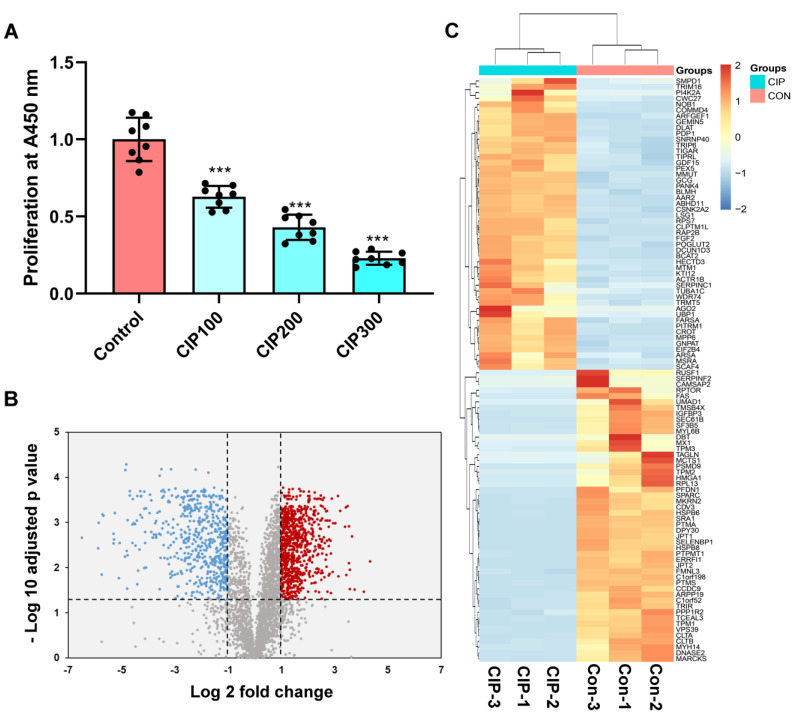
Identification of DEPs in CIP-stimulated VSMCs and cluster analysis. (**A**) CCK-8 results of VSMCs after CIP stimulation; (**B**) Volcano plot of DEPs; (**C**) Cluster analysis of the top 100 DEPs (50 upregulated, 50 downregulated). The symbols (***) represents *p* < 0.001 vs. control.

**Figure 2 jcm-12-01270-f002:**
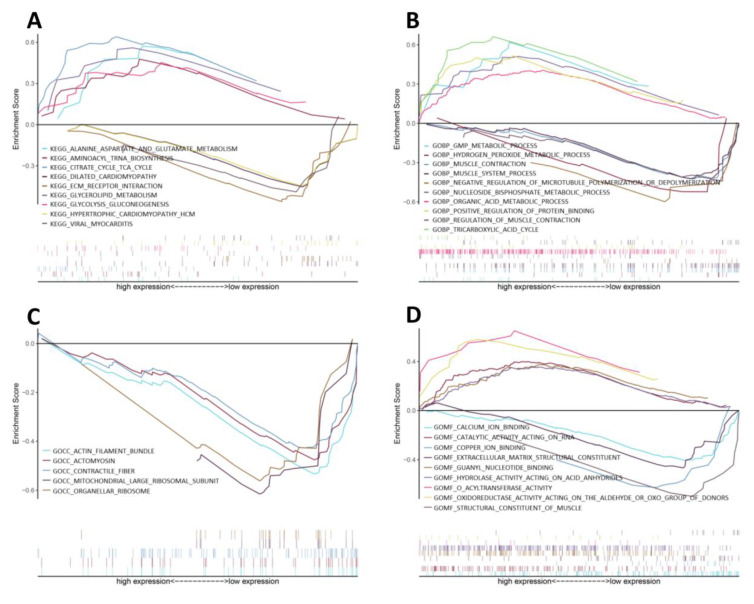
GSEA of CIP-stimulated VSMCs. GSEA results for the activation of (**A**) KEGG pathways, (**B**) biological processes, (**C**) cellular components and (**D**) molecular functions in CIP-stimulated VSMCs compared with control VSMCs.

**Figure 3 jcm-12-01270-f003:**
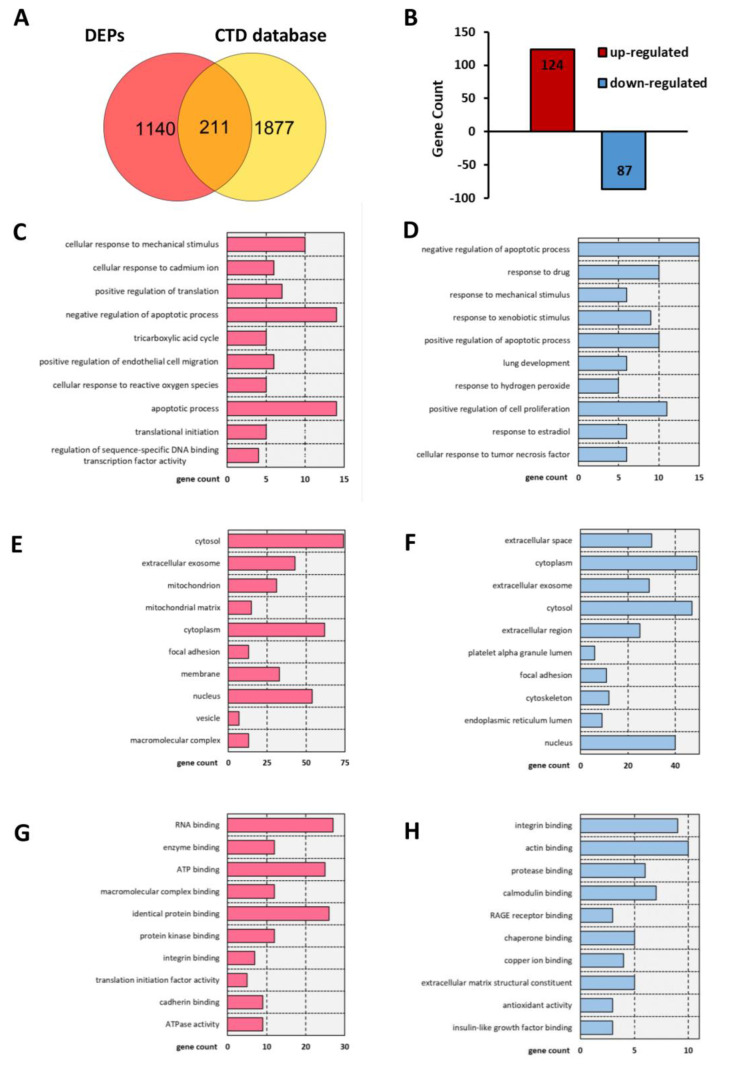
T Bar charts of the GO analysis results including BP, CC, and MF terms for 211 common genes. (**A**) Venn diagram showing the overlap of genes between DEPs and CTD-based AAD-related genes. (**B**) Bar charts of the up- and downregulated common molecules between DEPs and CTD-based AAD-related genes. (**C**,**D**) Enriched terms in the BP category for (**C**) upregulated and (**D**) downregulated common genes. (**E**,**F**) Enriched terms in the CC category for (**E**) up-regulated and (**F**) downregulated common genes. (**G**,**H**) Enriched items in the MF category for (**G**) upregulated and (**H**) downregulated common genes.

**Figure 4 jcm-12-01270-f004:**
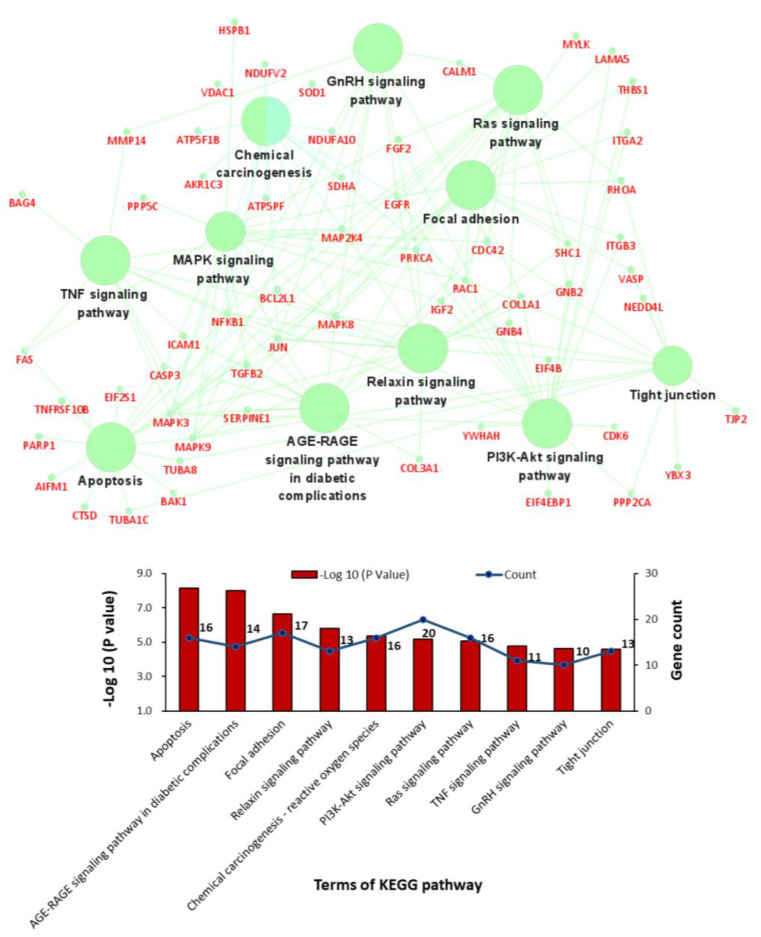
Bar charts and networks of the KEGG pathway enrichment analysis results.

**Figure 5 jcm-12-01270-f005:**
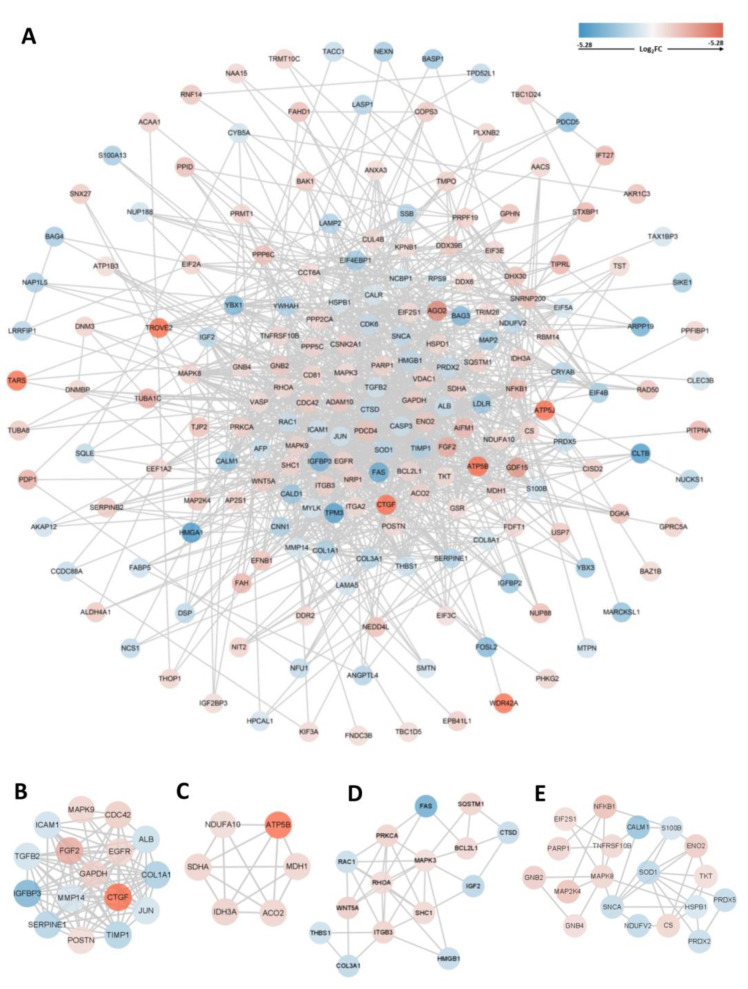
PPI network and module analysis. (**A**) PPI network of 211 common genes and (**B**–**E**) top four gene cluster modules.

**Figure 6 jcm-12-01270-f006:**
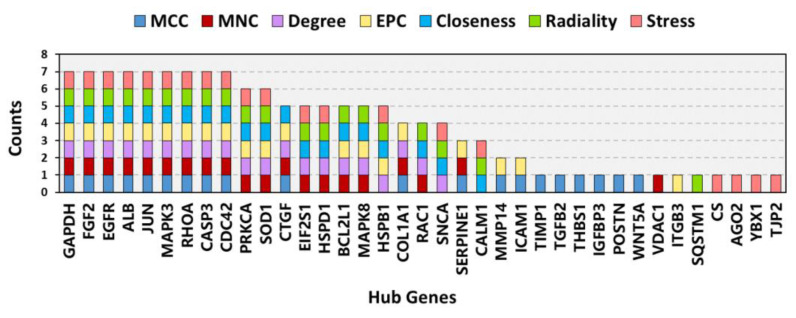
Stacked bar plots of the top 20 hub genes according to seven common algorithms conducted with the plugin cytoHubba.

**Figure 7 jcm-12-01270-f007:**
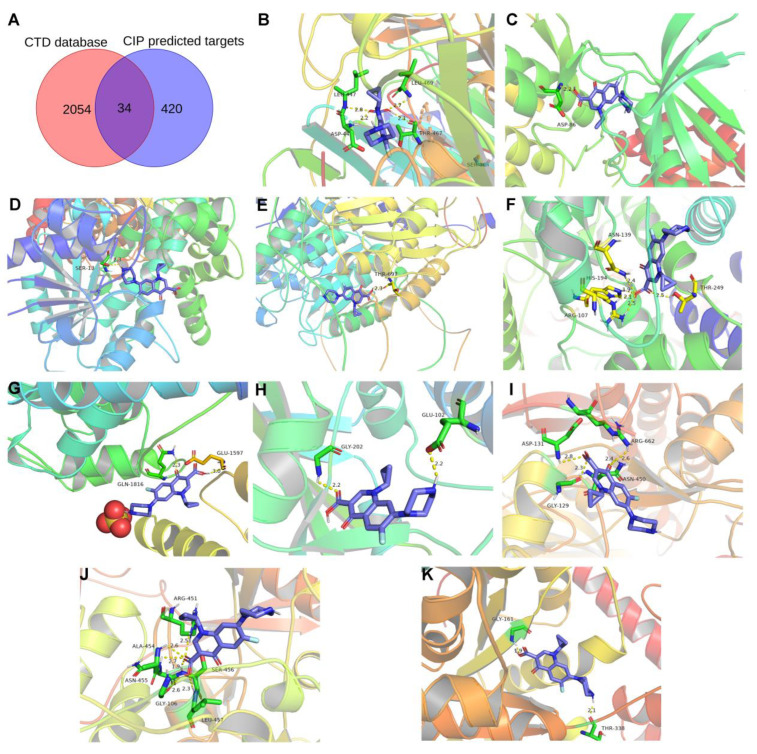
CIP target prediction and molecular docking. (**A**) Venn diagram showing the overlapping components among the predicted CIP targets and CTD-based AAD-related genes. (**B**–**K**) Molecular docking results for AOC3, CDK5R1, GPD1L, HDAC6, LDHB, MYO5A, PARP1, RAC1, SDHA, and WARS1.

**Figure 8 jcm-12-01270-f008:**
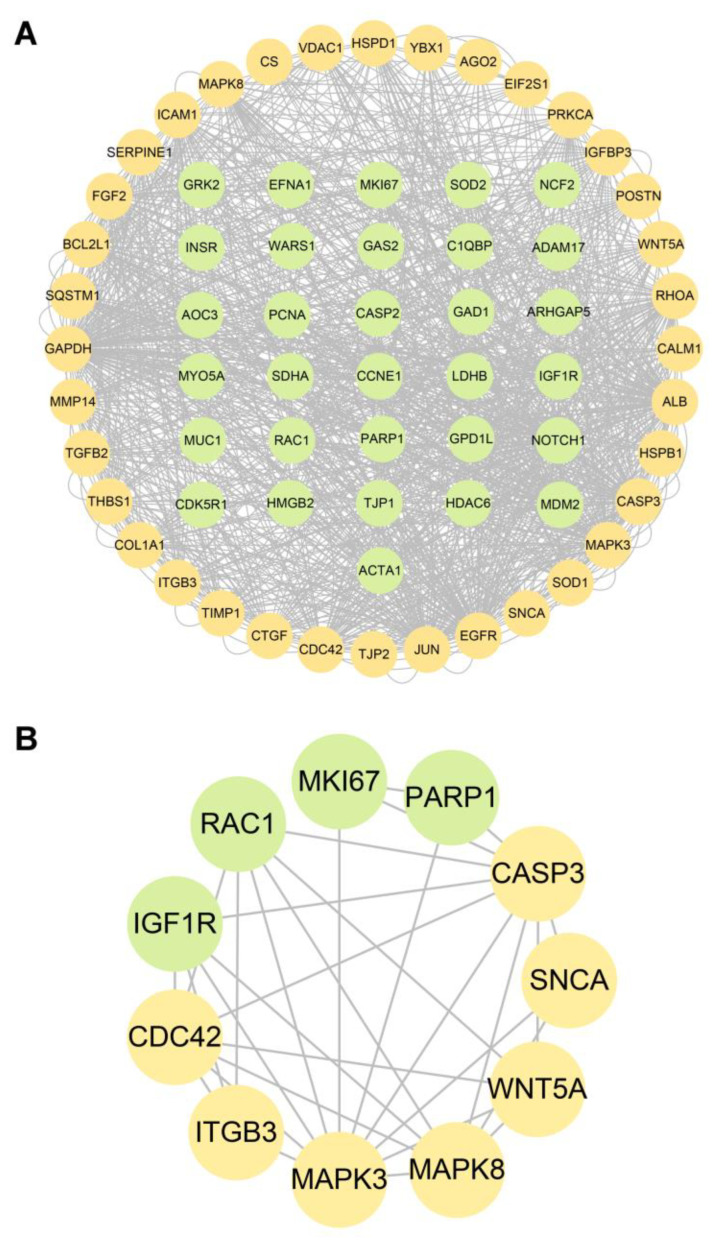
PPI network between AAD-related CIP targets and hub genes of CIP-stimulated DEPs. (**A**) PPI network. (**B**) Gene cluster module. Green nodes: AAD-related CIP targets. Yellow nodes: hub genes among the genes encoding CIP-stimulated DEPs.

**Table 1 jcm-12-01270-t001:** Details of the 34 intersecting target genes and their CIP binding energies.

Gene Symbol	Protein Names	Binding Energies of CIP (kcal/mol)	Expression in CIP Stimulated VSMCs	
			Log_2_FC	*p* Value
*RAC1*	Ras-related C3 botulinum toxin substrate 1	−8.2	−1.52	0.01
*GYG1*	Glycogenin-1	−7.9	−0.34	0.76
*C1QBP*	Complement component 1 Q subcomponent-binding protein, mitochondrial	−7.5	−0.19	0.07
*TJP1*	Tight junction protein ZO-1	−6.2	0.11	0.26
*SOD2*	Superoxide dismutase [Mn], mitochondrial	−6.2	0.27	0.10
*LDHB*	L-lactate dehydrogenase B chain	−8.8	0.57	0.00
*PCNA*	Proliferating cell nuclear antigen	−6.9	0.63	0.02
*WARS1*	Tryptophan--tRNA ligase, cytoplasmic	−8.0	0.68	0.01
*MYO5A*	Unconventional myosin-Va	−8.4	0.82	0.00
*ARHGAP5*	Rho GTPase-activating protein 5	−6.8	1.07	0.11
*PARP1*	Poly [ADP-ribose] polymerase 1	−9.1	1.33	0.00
*SDHA*	Succinate dehydrogenase [ubiquinone] flavoprotein subunit, mitochondrial	−8.2	1.42	0.00
*GPD1L*	Glycerol-3-phosphate dehydrogenase 1-like protein	−9.0	-	-
*AOC3*	Membrane primary amine oxidase	−8.6	-	-
*CDK5R1*	Cyclin-dependent kinase 5 activator 1	−8.4	-	-
*HDAC6*	Histone deacetylase 6	−8.4	-	-
*CCNE1*	G1/S-specific cyclin-E1	−7.5	-	-
*IGF1R*	Insulin-like growth factor 1 receptor	−7.5	-	-
*GRK2*	Beta-adrenergic receptor kinase 1	−7.4	-	-
*NCF2*	Neutrophil cytosol factor 2	−7.4	-	-
*DDX50*	ATP-dependent RNA helicase DDX50	−7.3	-	-
*MKI67*	Proliferation marker protein Ki-67	−7.3	-	-
*MUC1*	Mucin-1	−7.3	-	-
*CASP2*	Caspase-2	−7.2	-	-
*EFNA1*	Ephrin-A1	−7.2	-	-
*GAD1*	Glutamate decarboxylase 1	−7.2	-	-
*ADAM17*	Disintegrin and metalloproteinase domain-containing protein 17	−7.1	-	-
*ACTA1*	Actin, alpha skeletal muscle	−6.9	-	-
*GAS2*	Growth arrest-specific protein 2	−6.9	-	-
*INSR*	Insulin receptor	−6.9	-	-
*NOTCH1*	Neurogenic locus notch homolog protein 1	−6.8	-	-
*RUNX1T1*	Protein CBFA2T1	−6.8	-	-
*HMGB2*	High mobility group protein B2	−6.6	-	-
*MDM2*	E3 ubiquitin-protein ligase Mdm2	−5.7	-	-

## Data Availability

The data presented in this study are available on request from the corresponding author. The proteomics data have been deposited in the ProteomeXchange Consortium via the iProX partner repository with the dataset identifier PXD037691.

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
