# Peer review of "Investigation of the Pathogenic Mechanism of Ciprofloxacin in Aortic Aneurysm and Dissection by an Integrated Proteomics and Network Pharmacology Strategy"

_jcm, 2023, doi:10.3390/jcm12041270_

Round 1
Reviewer 1 Report
Review of the manuscript: “Investigation of the potential pathogenic mechanism of ciprofloxacin in aortic aneurysm and dissection by an integrated proteomics and network pharmacology strategy”
The manuscript describes the study upon human aortic smooth muscle cells (HASMC) – protein expression changes after ciprofloxacin stimulation assessed by the proteomic methods together with protein-protein interaction and target gene prediction. The background clinical problem investigated in this study is important – increased risk of aortic dissection and aneurysm development after treatment with ciprofloxacin – a widely used fluoroquinolone antibiotic. A big advantage of this study is approach to the problem with high-throughput methods from systemic biology – proteomics with additional tools like GSEA, Functional Enrichment Analysis, PPI, Target Gene Prediction. However, one and significant problem with the study is the use of high, supraphysiological concentrations of ciprofloxacin – 100 μg/ml, 200 μg/ml and 300 μg/ml, whereas peak concentrations of ciprofloxacin in human plasma is about 1.26 μg/ml (after administration of a single oral dose of 250 mg) [1]. Moreover, also in prostate tissue ciprofloxacin concentration is about 1.36 μg/g of prostate tissue (dosage – 200 mg 3 times a day for three days) [2]. This information should be provided and defined as a limitation of the study. It’s important for the clinicians who will be potential readers of this article.
Therefore I decided to recommend the major revision to the manuscript.
Apart from that, there are some minor flaws:
1. Line 55 – information about correlations between fluoroquinolones use and AAD in humans should be provided in more details (relative risks ecc).
2. Line 70 – “and provide new references for the prevention of AAD” – this fragment should be deleted, study concentrates upon pathomechanisms.
3. Figure 2, caption – abbreviations “CCs”, “MFs” and “BPs” should be explained in the text.
4. In the discussion section, investigations in which similar proteomic methods were applied should be briefly presented (at least one).
5. Limitations of the study should be also discussed.
6. There are some linguistic and typing errors, e.g. “cel-lular” in line 41; “AAD is a potentially life-threatening disease whose pathogenesis” in line 288 (should be – “(…) which pathogenesis”.
Nevertheless, the study is interesting, clinical problem – important and methodology – innovative. Therefore this manuscript is, according to me, hopeful for the publication in Journal of Clinical Medicie..
References
1. Sudo RT, Melo PA, Suarez-Kurtz G. Pharmacokinetics of oral ciprofloxacin in healthy, young Brazilian subjects. Braz J Med Biol Res. 1990;23(12):1315-21
2. Morita M, Nakagawa H, Suzuki K. [Ciprofloxacin concentration in human prostatic tissue following 3 days' administration]. Hinyokika Kiyo. 1991 May;37(5):563-6.
Author Response
Dear Reviewer,
Thanks very much for spending time reviewing our manuscript. Your comments are all valuable and very helpful for revising and improving our manuscript. We have revised the manuscript following your suggestions point to point, which are critically important to our work.
Comments and Suggestions for Authors
Review of the manuscript: “Investigation of the potential pathogenic mechanism of ciprofloxacin in aortic aneurysm and dissection by an integrated proteomics and network pharmacology strategy”. The manuscript describes the study upon human aortic smooth muscle cells (HASMC) – protein expression changes after ciprofloxacin stimulation assessed by the proteomic methods together with protein-protein interaction and target gene prediction. The background clinical problem investigated in this study is important – increased risk of aortic dissection and aneurysm development after treatment with ciprofloxacin – a widely used fluoroquinolone antibiotic. A big advantage of this study is approach to the problem with high-throughput methods from systemic biology – proteomics with additional tools like GSEA, Functional Enrichment Analysis, PPI, Target Gene Prediction. However, one and significant problem with the study is the use of high, supraphysiological concentrations of ciprofloxacin – 100 μg/ml, 200 μg/ml and 300 μg/ml, whereas peak concentrations of ciprofloxacin in human plasma is about 1.26 μg/ml (after administration of a single oral dose of 250 mg) [1]. Moreover, also in prostate tissue ciprofloxacin concentration is about 1.36 μg/g of prostate tissue (dosage – 200 mg 3 times a day for three days) [2]. This information should be provided and defined as a limitation of the study. It’s important for the clinicians who will be potential readers of this article. Therefore, I decided to recommend the major revision to the manuscript.
Response: Thanks for your good suggestion. The concentrations of ciprofloxacin used in our study were 100-300 μg/ml, which was consistent with several previous studies that explored the effect of ciprofloxacin on cellular function in vitro [1-2]. Even so, we agree with the Reviewer that there exists a gap between the drug concentration used in the in vitro experiment and the actual blood concentration. As the Reviewer suggested, we have added this limitation in the Discussion section for the reference of clinicians (Lines 354-359).
References:
[1] LeMaire, S.A.; Zhang, L.; Luo, W.; Ren, P.; Azares, A.R.; Wang, Y.; Zhang, C.; Coselli, J.S.; Shen, Y.H. Effect of Ciprofloxacin on Susceptibility to Aortic Dissection and Rupture in Mice. JAMA Surg 2018, 153, e181804
[2] Xiang, B.; Abudupataer, M.; Liu, G.; Zhou, X.; Liu, D.; Zhu, S.; Ming, Y.; Yin, X.; Yan, S.; Sun, Y.; Lai, H.; Wang, C.; Li, J.; Zhu, K. Ciprofloxacin exacerbates dysfunction of smooth muscle cells in a microphysiological model of thoracic aortic aneurysm. JCI Insight 2023, 8, e161729. doi: 10.1172/jci.insight.161729
Apart from that, there are some minor flaws:
Point 1: Line 55 – information about correlations between fluoroquinolones use and AAD in humans should be provided in more details (relative risks ecc).
Response 1: According to the Reviewer’s advice, we have enriched this information in the Introduction section (Lines 53-59).
‘However, many studies have indicated that FNs are associated with an increased incidence of AAD. According to different population-based studies, FN use increases the risk of AAD with a hazard ratio from 1.31 to 2.43 [7-10]. Moreover, FN exposure was reported to be associated with an increased risk of adverse outcomes in AAD patients [11]. According to the Food and Drug Administration (FDA), FNs should not be used by patients with high-risk conditions, such as those with Marfan syndrome or aortic diseases [12].’
Point 2. Line 70 – “and provide new references for the prevention of AAD” – this fragment should be deleted, study concentrates upon pathomechanisms.
Response 2: As the Reviewer suggested, we have deleted this expression in the Abstract and Introduction section (Line 77).
Point 3. Figure 2, caption – abbreviations “CCs”, “MFs” and “BPs” should be explained in the text.
Response 3: Thanks for your suggestion. We have changed the abbreviations to full names in the figure caption (Line 204).
Figure 2. GSEA of CIP-stimulated VSMCs. GSEA results for the activation of (A) KEGG pathways, (B) biological processes, (C) cellular components and (D) molecular functions in CIP-stimulated VSMCs compared with control VSMCs.
Point 4. In the discussion section, investigations in which similar proteomic methods were applied should be briefly presented (at least one).
Response 4: As the Reviewer suggested, we have added the related information in the discussion section (Lines 347-349).
‘Meanwhile, a novel DIA-based proteomics technology promises robust and accurate quantification of proteins, which is increasingly being applied in cardiovascular disease research [53-54].’
Point 5. Limitations of the study should be also discussed.
Response 5: Thanks for your good suggestion. We have added the limitations of this study in the discussion (Lines 349-361).
‘Our findings have important clinical implications, but several limitations of this study need to be mentioned. First, although VSMCs remain the most important cellular components in the pathological process of AAD, other cells, such as vascular endotheliocytes and macrophages, also participate in the development of AAD. This study only focused on VSMCs, and the conclusion may not be extended to other cellular components. Second, consistent with previous studies that explored the effect of ciprofloxacin on cellular function in vitro [55-56], the concentrations of CIP used in our study were 100-300 μg/ml. However, the peak concentration of CIP in human plasma is approximately 1.26 μg/ml after administration of a single oral dose of 250 mg [57]. The gap between the drug concentration used in the in vitro experiment and the actual blood concentration may affect the interpretation of the current results. Third, all the data we analysed are from in vitro experiments and online databases, and confirmation via in vivo experiments will be needed in the future.’
Point 6. There are some linguistic and typing errors, e.g. “cel-lular” in line 41; “AAD is a potentially life-threatening disease whose pathogenesis” in line 288 (should be – “(…) which pathogenesis”.
Response 6: As the Reviewer suggested, we have revised the manuscript and checked the whole manuscript carefully. The language in the manuscript was also edited by a professional native speaker. Thanks for your patience and careful review.
We earnestly appreciate your work and hope that the revised manuscript will meet with your approval. Once again, thanks very much for your comments and suggestions.

Reviewer 2 Report
The manuscript entitled “Investigation of the potential pathogenic mechanism of ciprofloxacin in aortic aneurysm and dissection by an integrated proteomics and network pharmacology strategy " investigates the potential functional mechanism and molecular targets of fluoroquinolones in relation to AAD by an integrated proteomic and network pharmacology strategy. A total of 1351 differentially expressed proteins were identified in human aortic vascular smooth muscle cells (VSMCs) after ciprofloxacin (CIP) stimulation. However, the presented article could be published in Journal of clinical medicine after revision for the following.
1. Introduction part should be enriched with a sufficient background as well as a justification for the current research purpose.
2. Limitations of the study should be added.
3. Finally, English should be polished throughout the text.
Author Response
Dear Reviewer,
We would like to thank you for your kind letter and professional assessment of our manuscript. We have tried our best to revise our manuscript according to the comments and re-submit our manuscript in the system. Point-by-point responses to the recommendations are given below.
Comments and Suggestions for Authors
The manuscript entitled “Investigation of the potential pathogenic mechanism of ciprofloxacin in aortic aneurysm and dissection by an integrated proteomics and network pharmacology strategy " investigates the potential functional mechanism and molecular targets of fluoroquinolones in relation to AAD by an integrated proteomic and network pharmacology strategy. A total of 1351 differentially expressed proteins were identified in human aortic vascular smooth muscle cells (VSMCs) after ciprofloxacin (CIP) stimulation. However, the presented article could be published in Journal of clinical medicine after revision for the following.
Point 1. Introduction part should be enriched with a sufficient background as well as a justification for the current research purpose.
Response 1: Thanks for your good suggestion. We enriched the Introduction section according to the reviewer’s advice, as follows.
‘However, many studies have indicated that FNs are associated with an increased incidence of AAD. According to different population-based studies, FN use increases the risk of AAD with a hazard ratio from 1.31 to 2.43 [7-10]. Moreover, FN exposure was reported to be associated with an increased risk of adverse outcomes in AAD patients [11]. According to the Food and Drug Administration (FDA), FNs should not be used by patients with high-risk conditions, such as those with Marfan syndrome or aortic diseases [12].’ (Lines 53-59).
‘Evidence has shown that loss and dysfunction of VSMCs is associated with the occurrence and development of AAD and an increased risk of AAD in FN use. Thus, we hypothesized that CIP might affect the proliferation and survival of VSMCs by certain mechanisms.’ (Lines 69-72).
Point 2. Limitations of the study should be added.
Response 2: Thanks for your good suggestion. We have added the limitations of this study in the discussion (Lines 349-361).
Our findings have important clinical implications, but several limitations of this study need to be mentioned. First, although VSMCs remain the most important cellular components in the pathological process of AAD, other cells, such as vascular endotheliocytes and macrophages, also participate in the development of AAD. This study only focused on VSMCs, and the conclusion may not be extended to other cellular components. Second, consistent with previous studies that explored the effect of ciprofloxacin on cellular function in vitro [55-56], the concentrations of CIP used in our study were 100-300 μg/ml. However, the peak concentration of CIP in human plasma is approximately 1.26 μg/ml after administration of a single oral dose of 250 mg [57]. The gap between the drug concentration used in the in vitro experiment and the actual blood concentration may affect the interpretation of the current results. Third, all the data we analysed are from in vitro experiments and online databases, and confirmation via in vivo experiments will be needed in the future.
Point 3. Finally, English should be polished throughout the text.
Response 3: As the Reviewer suggested, we have polished the language in our manuscript by a professional native speaker. Thanks for your patience and careful review.
Thank you very much for spending the time reviewing our manuscript. We earnestly appreciate your work and hope that the revised manuscript will meet with your approval.

Round 2
Reviewer 1 Report
The flaws that I indicated in the review were corrected, therefore now in my opinion the paper is appropriate for publication.